# PRR11 in Malignancies: Biological Activities and Targeted Therapies

**DOI:** 10.3390/biom12121800

**Published:** 2022-12-01

**Authors:** Wei Han, Liang Chen

**Affiliations:** 1Department of Neurosurgery, Huashan Hospital, Shanghai Medical College, Fudan University, Shanghai 200040, China; 2National Center for Neurological Disorders, Shanghai 200040, China; 3Shanghai Key Laboratory of Brain Function and Restoration and Neural Regeneration, Shanghai 200040, China; 4Neurosurgical Institute, Fudan University, Shanghai 200040, China; 5Shanghai Clinical Medical Center of Neurosurgery, Shanghai 200040, China; 6State Key Laboratory of Medical Neurobiology, Institute for Translational Brain Research, MOE Frontiers Center for Brain Science, Fudan University, Shanghai 200032, China

**Keywords:** PRR11, malignancies, functional activities, signaling pathways, targeted therapy

## Abstract

Proline rich 11 (PRR11), initially renowned for its relevance with cell-cycle progression, is a proline-rich protein coding gene in chromosome 17q22-23. Currently, accumulating studies have demonstrated that PRR11 plays a critical role in cellular proliferation, colony formation, migration, invasion, cell-cycle progression, apoptosis, autophagy and chemotherapy resistance via multiple signaling pathways and biological molecules in several solid tumors. In particular, PRR11 also serves as a promising prognostic indicator in a limited number of human cancers, gradually manifesting its potential application for targeted therapies. In this review, we summarize functional activities, related signaling pathways and biological molecules of PRR11 in various malignancies and generalize potential application of PRR11 for targeted therapies, thereby contributing to further exploration of PRR11 in cancer treatment.

## 1. Introduction

Proline rich 11 (PRR11), situated in chromosome 17q22-23, is a proline-rich protein coding gene with ten exons and nine introns [1,2]. It is adjacent to spindle and kinetochore associated 2 (SKA2), sharing a NF-Y-regulated bidirectional promoter, which could be precisely modulated by p53 [3]. The PRR11 mRNA is 2408 bp in length, containing an open reading frame encoding 360 amino acids [2]. At the protein level, PRR11 is composed of a nuclear localization signal (NLS), two proline rich regions (PRs) and a zinc finger domain (ZFD) [4] (Figure 1a,b).

Intriguingly, PRR11 expression is significantly upregulated in various human cancers, except acute myeloid leukemia (LAML), per the gene expression omnibus (GEO) database (Figure 1c). With regard to its biological function, PRR11 could be initially prominent for its correlation with the cell cycle of cancer cells [2]. Subsequent experiments have validated that PRR11 is involved in multiple cellular activities [5,6,7]. For instance, PRR11 extensively facilitates the proliferation, invasion and cell-cycle progression of tongue squamous cell carcinoma (TSCC) cells [8]. In non-small cell lung cancer (NSCLC) cells, cellular apoptosis and autophagy would be reduced by PRR11 [9,10]. More significantly, in vivo xenograft tumor growth, deriving from ovarian cancer cells, is accelerated by PRR11 [11]. Additionally, prognostic significance and potential application for targeted therapies of PRR11 in several solid tumors also have been verified [12,13,14]. Therefore, it is imperative to summarize progress in exploring biological and clinical roles of PRR11.

In this review, we try to generalize functional activities, related signaling pathways and biological molecules of PRR11 in malignancies and the potential application of PRR11 for targeted therapies, thus contributing to further exploration of PRR11 in cancer treatment (Figure 2 and Figure 3 and Table 1 and Table 2).

## 2. Oncogenic role of PRR11 in Various Malignancies

### 2.1. Tongue Squamous Cell Carcinoma (TSCC)

TSCC, identified as the most common type of tongue cancers, has a propensity of regional recurrence and lymphoid metastasis [23,24]. Though the standardized chemotherapy of cisplatin for TSCC patients would bring survival amelioration, the emergence of chemo-resistance is inevitable, ultimately leading to functional defects [24,25]. Thus, reliable therapeutic targets are essential for improvement of clinical efficacy in TSCC patients. According to assessment by qRT-PCR and Western blot, PRR11 expression is dramatically augmented in TSCC specimens and several TSCC cell lines, including Cal27, SCC15, HSC3, HSC4, HSC6, UM1 and UM2, relative to the normal oral keratinocytes (NOKs) and equivalent non-cancerous tissues [8,22]. Given the clinical characteristics and immunohistochemistry of these TSCC patients, PRR11 serves as an independent prognostic factor for TSCC patients via correlation with the clinical stage, TNM classification, vital status and overall survival [22]. Subsequent in vitro experiments illustrated that PRR11 overexpression significantly enhanced cellular proliferation, invasion and cell-cycle progression through downregulation of p21 and p27 and upregulation of cyclin-dependent kinase 2 (CDK2) and cyclin A2 in TSCC cells. In addition, an in vivo subcutaneous tumorigenicity assay also found that PRR11 knockdown diminished the tumor size and Ki-67 expression in TSCC tissues [8]. Above all, PRR11 serves as an oncogene in TSCC patients, which could be utilized for targeted therapies.

### 2.2. Esophageal Squamous Cell Carcinoma (ESCC)

ESCC, accounting for 90% of esophageal carcinoma, has the highest occurrence in East Asian countries [26,27]. Due to its metastasis, recurrence and lack of comprehensive treatment, the overall five-year survival rate of ESCC patients is still dismal, with a proportion of roughly 15% [26,28]. Therefore, exploring novel diagnostic indicators would be of clinical benefit to ESCC patients. Analysis of ESCC patients from the GEO database has verified the oncogenic role of PRR11 in ESCC tumorigenesis [29]. Relatively higher expression levels of PRR11 are also detected in 12 ESCC cell lines and 38 ESCC tissues than in 2 normal esophageal epithelial cells (NEECs) and the matched adjacent non-tumor tissues [4,15]. Further investigation revealed that PRR11 dramatically facilitated cellular proliferation and the migratory and invasive capacities of ESCC cells targeting Akt and EMT signaling via collaboration with SKA2 [4]. Aiming at cancer stem cells (CSCs), PRR11 knockdown reduced CSC-like phenotypes and tumorigenicity of ESCC cells in vitro and in vivo through Wnt/β-catenin signaling [15]. Surprisingly, chalcone, with chemical modification by dithiocarbamate scaffolds, suppressed the cell growth, migration and invasion of ESCC cells targeting PRR11 [30], showing great potential application for targeted therapy of ESCC patients. However, there is no evidence in evaluating the connection between PRR11 expression and clinical features of ESCC patients. In summary, PRR11 might represent a novel prognostic marker for ESCC patients, which needs further clinical verification.

### 2.3. Non-small Cell Lung Cancer (NSCLC)

NSCLC, the most common histopathological classification of lung cancer, has become the main cause of cancer-associated deaths in urban populations [31]. Though comprehensive treatment has developed in recent years, earlier metastasis usually occurs when initially diagnosed, leading to poor prognosis [32,33]. Thereby, there is an urgent demand to develop novel targets for NSCLC patients. Compared to 8 normal tissues, the expression level of PRR11 in 40 NSCLC tissues is obviously upregulated. Further investigation concerning clinical features of NSCLC patients revealed that PRR11 contributes to an improved clinical stage and shorter overall survival [2,9,34]. For cellular activities, PRR11 was involved in proliferation, migration, cell-cycle progression, invasion, apoptosis and autophagy of NSCLC cells [2,9,10]. Similarly, in vivo tumorigenicity could also be modulated by PRR11 [2,9]. Functionally, PRR11 activated phosphorylation of Akt and mTOR and recruited the actin-related protein 2/3 complex, thus activating Akt/mTOR signaling and F-actin assembly [10,35]. Moreover, a complex, regulated by p53, consisting of PRR11 and SKA2 contributed to the progression of NSCLC cells [3,36]. As for the transcription level, PRR11 targeted the E2F1/pituitary tumor-transforming gene 1 (PTTG1) axis with involvement of NSCLC progression [16]. Preliminary studies on treating NSCLC patients demonstrated that lncRNA DLX6-AS1 knockdown inhibited cell proliferation, migration, invasion and initiated apoptosis targeting PRR11 in NSCLCs [37]. Collectively, PRR11 could be a reliable prognostic factor and therapeutic target for NSCLC patients.

### 2.4. Breast Cancer (BRCA)

BRCA, the most common solid tumor among women, is the second cause of tumor-related death in women [38]. BRCA has four categories, including estrogen receptor (ER^+^), progesterone receptor (PR^+^), human epidermal receptor 2 (HER2^+^) and triple negative breast cancer (TNBC), which correlate to the expression of hormonal receptors [39]. However, the progress in early diagnosis and treatment for BRCA patients is still far from satisfactory, appealing to more reliable biomarkers. In 109 BRCA patients and 260 ER^+^ BRCA patients, PRR11 promotes more lymph node metastasis, higher Ki-67 ratio and endocrine resistance, thus diminishing the overall survival, recurrence-free survival and relapse-free survival [1,5,40]. In cellular activities, PRR11 facilitated cellular proliferation, migration, invasion and antiestrogen resistance [1,5,17,18]. Subsequent functional investigation uncovered that PRR11 reduced the expression of E-cadherin, cytokeratin-18 and upregulated the expression of N-cadherin, vimentin and fibronectin targeting EMT-inducing transcription factors Snail, Slug, zinc finger E-box binding homeobox 1 (ZEB1) and ZEB2 [5]. Moreover, PRR11 blocked p85 homodimerization and sensitized to ligand-induced PI3K activation, suggesting that PRR11 amplification confers resistance to estrogen deprivation through hyperactivation of the PI3K pathway [1]. Notably, ultrasonic irradiation and SonoVue microbubbles-mediated RNA interference targeting PRR11 could exert anti-cancer effects via PRR11 [41]. However, the oncogenic role of PRR11 in vivo is still blurred, demanding additional exploration. In brief, PRR11 acts as a prognostic factor for BRCA patients, which could also be a reliable therapeutic target.

### 2.5. Gastric Cancer (GC) and Colorectal Cancer (CRC)

GC, originating from the epithelium of the gastric mucosa, evidently has regional differences [42]. Disappointingly, the early diagnosis rate of GC patients is still negligible, contributing to more distant metastasis and lymph node metastasis [43]. Accordingly, effective diagnostic indicators are pivotal to predict the outcomes of GC patients and facilitate the development of targeted therapy. In 216 GC patients, PRR11 expression is positively correlated with TNM stage and tumor differentiation, accompanied by the shorter overall survival of these patients. Further exploration on cellular activities illustrated that PRR11 overexpression promoted cellular proliferation and colony formation of GC cells [13]. As the vital cell subsets in tumorigenesis, GC stem cells were also modulated by PRR11 in self-renewal and maintaining stemness [6]. Moreover, in vivo tumorigenicity was suppressed by PRR11 in tumor volume and weight. Functionally, PRR11 exerted oncogenic effects mainly upregulating collagen triple helix repeat containing 1 (CTHRC1), downregulating latexin (LXN) or targeting MAPK signaling [6,13].

Nowadays, CRC is the fourth most deadly cancer, with low diagnostic rates and approximately 900,000 deaths annually [44]. Though organoids could provide long-term cultures of normal intestinal epithelial cells and good animal models for CRC, the prognosis of CRC patients is still poor [45]. Compared to normal ones, PRR11 is obviously elevated in CRC tissues and cells. Exploration of its biological activities showed that PRR11 silencing suppressed the proliferation, invasion and migration of CRC cells and xenograft tumor growth in vivo. Mechanically, PRR11 knockdown suppressed the EGFR/ERK/AKT pathway via inhibiting CTHRC1 expression [19]. However, further survival analysis of CRC patients targeting PRR11 expression is still in urgent demand. In summary, PRR11 might have an oncogenic role in the progression and development of GC and CRC patients.

### 2.6. Hepatocellular Carcinoma (HCC) and Hilar Cholangiocarcinoma (HCCA)

HCC, characterized with high morbidity, is the most common primary liver cancer in humans [46]. Though surgical resection and adjuvant chemotherapy of sorafenib would improve overall survival of HCC patients, the incidence of HCC and the mortality of HCC patients were still high [47,48]. Due to the lack of explicit clinical symptoms in the early stage, effective diagnostic biomarkers are in urgent need. Analysis of the weighted gene co-expression network in HCC found that PRR11 is positively connected to the progression and prognosis of HCC patients [49]. Further exploration on clinical characteristics of 80 pairs of HCC tissues and adjacent non-tumor liver tissues confirmed that PRR11 overexpression contributes to larger tumor size, improved TNM stage and shorter overall survival. In cellular activities, PRR11 silencing inhibited cellular proliferation, migration and invasion in vitro. Similar inhibitory effects on xenograft tumor growth of HCC cells were also observed in tumor volume, weight and Ki-67 ratio. Functionally, PRRl1 knockdown reduced the expression of N-cadherin while increasing E-cadherin expression, thus inactivating the EMT process. Furthermore, PRR11 was positively correlated with β-catenin, c-myc and cyclin D1 targeting β-catenin signaling [20]. Of course, PRR11 targeted the E2F1/PTTG1 axis to modulate gene transcription of HCC cells [16].

HCCA, also known as Klatskin tumor, is a common malignant tumor of the biliary system [50]. Despite current advances in imaging and surgical techniques, the prognosis of HCCA patients is still poor due to its special location, aggressive growth and adjacent relationship with the hilar [51]. In order to facilitate early diagnosis, novel biomarkers should be identified for HCCA. In 49 patients with HCCAs, PRR11 overexpression is linked to earlier invasion, more lymph node metastasis and shorter overall survival. For cellular activities, PRR11 knockdown suppressed hilar cholangiocarcinoma cell proliferation, migration, cell-cycle progression and tumor growth in vitro and in vivo. Subsequent functional analysis illustrated that PRR11 increased the expression of vimentin (VIM) protein and decreased expression of E-cadherin, thus initiating the EMT process. There were also some other downstream regulatory proteins, including ubiquitin carboxyl-terminal hydrolase 1 (UCHL1), early growth response protein (EGR1) and system a amino acid transporter 1 (SNAT1) [12]. There is also a demand for additional experiments on cellular apoptosis and autophagy in HCC and HCCA cells. Consequently, PRR11 plays an oncogenic role in the progression of HCC and HCCA, which might be a valuable prognostic marker and therapeutic target.

### 2.7. Pancreatic Cancer

Pancreatic cancer, which exhibits extensive metabolic reprogramming, is one of the malignant tumors of the digestive tract [52]. The clinical features of pancreatic cancer, including late diagnosis, early metastasis and limited reaction to chemotherapy or radiotherapy, lead to a poor five-year survival rate, which is around 10% [53]. To improve early diagnosis and overall survival, new biomarkers should be further explored. The expression level of PRR11 in patients with pancreatic cancers was significantly higher than those of normal patients [14]. Analysis of clinical parameters of 38 pancreatic cancer samples and 10 normal pancreatic tissues demonstrated that PRR11 overexpression is positively correlated with tumor invasion and differentiation, accompanied by shorter overall survival [14,54]. In vitro, PRR11 knockdown markedly inhibited cellular migration in pancreatic cancer cells [14]. Mechanically, the E2F1/PTTG1 axis was regulated by PRR11 to participate in the progression of PAAD cells [16]. Notably, further exploration of treating pancreatic cancers revealed that ubiquitin specific protease 34 (USP34) knockdown inhibited proliferation and migration, and induced apoptosis targeting PRR11 [55]. Moreover, miR-144-3p decreased PRR11 expression to oppress cell proliferation and initiate cell apoptosis and cell cycle arrest [56]. In summary, PRR11 might be a reliable prognostic factor for pancreatic cancer patients, for which the biological interventions should be further validated.

### 2.8. Ovarian Cancer

Ovarian cancer, the third most common gynecologic malignancy, mainly caused deaths in patients with gynecological tumors [57]. Despite the progress in comprehensive treatment, 60–70% of ovarian cancer patients are usually in the advanced clinical stage, leading to poor prognosis [58,59]. Hence, reliable indicators for diagnosis and treatment of ovarian cancer patients are indispensable. Compared to normal ovarian surface epithelium tissues and a normal ovarian epithelial cell line IOSE80, PRR11 expression is significantly upregulated in 51 pairs of ovarian cancer tissues and 4 ovarian cancer cell lines, including Caov3, SKOV3, OVCAR3 and HO-8910 [7]. Concluding from the clinical features of 51 pairs of ovarian cancer patients and 49 primary invasive ovarian cancer patients, PRR11 is positively correlated with improved FIGO stage, larger tumor size, more lymph node metastasis and shorter overall survival [7,11]. For cellular activities, cell proliferation, migration and invasion were involved in the biological functions of PRR11 [7]. In addition, PRR11 obviously promoted xenograft tumor growth in vivo [11]. Mechanically, PRR11 targeted c-myc, cyclin D1, matrix metallopeptidase 2 (MMP2), tissue inhibitor of metalloproteases 2 (TIMP-2), EGR1 and N-cadherin to exert the oncogenic effects on cell proliferation and migration [7,11]. PRR11 also increased the expression of p-Akt/Akt and cytoplasmic and nuclear β-catenin, thus activating PI3K/Akt/β-catenin signaling [7]. However, there is still a demand for the investigation of PRR11 activities on cell cycle, apoptosis and autophagy. To summarize, PRR11 could be a reliable prognostic factor for ovarian cancer patients.

### 2.9. Osteosarcoma

Osteosarcoma is the most prevalent and malignant bone cancer in children and adolescents [60]. With dramatic improvement in chemotherapy and surgery, the long-term survival rates of osteosarcoma patients are only 70% due to its variable distribution, distant metastases and lack of clinical symptoms [61]. Therefore, reliable biomarkers for osteosarcoma patients are in urgent demand. Compared with paracancerous tissues, PRR11 expression is elevated in 62 cases of osteosarcoma tissues. In further analysis of PRR11 expression in osteosarcoma patients, high PRR11 expression is correlated with larger tumor size, advanced Enneking stage and lymph node metastasis. Relative higher expression of PRR11 was also observed in osteosarcoma cell lines, including SAOS2, MG63 and U2OS. In cellular activities, PRR11 was involved in affecting proliferation, migration, invasion and apoptosis. Mechanically, PRR11 knockdown decreased β-catenin and p-GSK-3β and increased p-β-catenin and GSK-3β, thus inactivating Wnt/β-catenin signaling. In addition, PRR11 knockdown also increased the expression of E-cadherin, but reduced vimentin and fibronectin in the EMT signaling [21]. Above all, PRR11 might be an independent and reliable biomarker and therapeutic target for osteosarcoma patients.

## 3. PRR11 Involved Oncogenic Signaling Pathways and Biological Molecules

### 3.1. Signaling Pathways

Gathering evidence has confirmed that several signaling pathways and biological molecules have participated in the progression and development of human cancers [62,63,64,65]. Currently, main signaling pathways, including EMT, MAPK, PI3K/Akt/mTOR and Wnt/β-catenin signalings, are involved in the biological activities of PRR11. In EMT signaling, PRRl1 overexpression increases the expression of N-cadherin, vimentin and fibronectin, but reduces E-cadherin and cytokeratin-18 expression, thus activating EMT signaling [4,5,20]. Subsequently, some downstream regulatory proteins, such as UCHL1, EGR1 and SNAT1, are also activated by PRR11 [12]. Furthermore, activities of EMT-inducing transcription factors, including Snail, Slug, ZEB1 and ZEB2, could also be detected in the functional processes of PRR11 [5]. With regard to PI3K/Akt/mTOR signaling, PRR11 blocks p85 homodimerization and sensitizes to ligand-induced PI3K activation [1]. PRR11 also activates the phosphorylation of Akt and mTOR, thus activating Akt/mTOR signaling [10]. In Wnt/β-catenin signaling, PRR11 significantly enhances the activity of TOP/FOP flash luciferase reporter [15]. Evidently, cytoplasmic and nuclear β-catenin could also be upregulated by PRR11, thereby promoting the transcription of c-myc and cyclin D1 [7,20].

### 3.2. Biological Molecules

In addition to the main signaling pathways mentioned above, several biological molecules are also modulated by PRR11. For cellular proliferation and migration, PRR11 targets MMP2 and TIMP-2 to exert oncogenic effects [7]. In the meanwhile, cell-cycle progression is significantly modulated by PRR11 via downregulation of p21 and p27 and upregulation of CDK2 and cyclin A2 [8]. Moreover, CTHRC1, LXN, EGR1, UCHL1, SNAT1 and PTTG1 are also reliable downstream targets for PRR11 [11,12,13,16]. Above all, EMT, PI3K/AKT/mTOR, Wnt/β-catenin signalings and biological molecules, including MMP2, TIMP-2, p21, p27, CDK2, cyclin A2, CTHRC1, LXN, EGR1, UCHL1, SNAT1 and PTTG1, have been testified to participate in the functional processes of PRR11, which might be feasible approaches for the targeted therapy of PRR11 in human cancers (Figure 2).

## 4. Biological and Clinical Significance of PRR11 in Human Cancers

### 4.1. PRR11 Involved in Tumorigenesis of Human Cancers

Cancer stem cells, considered as a vital cell cluster in cellular proliferation, drug resistance and tumor recurrence, are equipped with self-renewal and multi-directional differentiation [66]. In ESCC, BRCA and GC, PRR11 facilitated the self-renewal and stemness of stem cells, contributing to drug resistance [1,6,30]. Subsequently, in vitro and in vivo models were utilized for evaluation of PRR11 in tumorigenesis. In vitro, cellular proliferation and cell-cycle progression were evidently boosted by PRR11 in TSCC, ESCC, NSCLC, BRCA, GC, CRC, HCC, HCCA, Pancreatic cancer, Ovarian cancer and Osteosarcoma [4,5,7,8,10,12,13,14,16,19,21]. Similarly, in in vivo models, PRR11 promoted xenograft tumor growth in tumor size, weight and Ki-67 expression in TSCC, NSCLC, GC, CRC, HCC and Ovarian cancer [2,6,8,9,11,13,19,20]. Hence, PRR11 might be great of biological significance in tumorigenesis of human cancers.

### 4.2. PRR11 Involved in Diagnosis and Prognosis of Malignancies

Compared with non-tumorous tissues, PRR11 is evidently upregulated in tumor tissues, including ESCC, NSCLC, CRC, HCC, Pancreatic cancer, Ovarian cancer and Osteosarcoma, which would contribute to the improvement of clinical diagnosis in these cancers [2,4,7,9,14,15,19,20,21,34]. More importantly, PRR11 also serves as a prognostic biomarker for several human cancers. Considering the clinical features and prognosis of these patients, PRR11 is positively correlated with earlier tumor invasion, advanced clinical stage, TNM classification, more lymph node metastasis and higher Ki-67 ratio, which usually are accompanied by shorter overall survival in TSCC, NSCLC, BRCA, GC, HCC, Pancreatic cancer, Ovarian cancer and Osteosarcoma [1,5,7,11,12,13,14,20,21,22,40,54]. Therefore, PRR11 acts as a promising prognostic indicator in a limited number of human cancers, gradually manifesting its potential application for targeted therapies (Table 1 and Table 2).

## 5. PRR11 Involved Targeted Therapies in Various Malignancies

In recent years, targeted therapies have gradually exhibited potential application in cancer treatment with extraordinary clinical efficacy [67,68]. Based on these investigational results, targeted therapies of PRR11, especially interactions between PRR11 and miRNAs, lncRNAs and other molecules, would be utilized for cancer treatment.

### 5.1. Interactions between PRR11 and miRNAs

The miRNAs, a group of non-coding RNAs, play vital roles in the initiation and progression of human cancers [69]. In most circumstances, miRNAs induce the degradation and translational repression of target mRNAs to modulate gene expression targeting the 3′ untranslated region (3′ UTR) of mRNAs [70]. For instance, the luciferase activity of the PRR11 3-UTR wild-type reporter gene, not the PRR11 3-UTR mutated reporter gene, was significantly decreased by miRNA-26b-5p, thus suppressing PRR11 expression [71]. Similarly, in prostate cancer, osteosarcoma, pancreatic cancer and breast cancer, miR-195, miRNA-211-5p, miR-204-5p and miR-144-3p directly targeted PRR11 3′-UTR to participate in cellular activities, including proliferation, migration, cell cycle, apoptosis and angiogenesis [18,56,72,73]. Therefore, exploring more miRNAs targeting PRR11 3′-UTR might be a reliable approach to regulate PRR11 expression in cancer treatment.

### 5.2. Interactions between PRR11 and lncRNAs

LncRNAs usually modulate gene expression by functioning as competing endogenous RNAs (ceRNAs), which sequester miRNAs and therefore protect their target mRNAs from repression [74]. As mentioned above, lncRNA DLX6-AS1 served as a ceRNA of miR-144 in NSCLC cells to upregulate PRR11 expression [37]. In particular, lncRNA CCDC26 elevated circRNA_ANKIB1 expression, acting as a ceRNA of miR-195-5p, to upregulate PRR11 expression in myeloid leukemia cells [75]. However, the biological mechanism of lncRNA AC099850.3 as an oncogene in hepatocellular carcinoma via PRR11/PI3K/AKT axis is still uncertain [76].

### 5.3. Interactions between PRR11 and other Molecules

Preliminary experiments found that SonoVue microbubbles-mediated RNA interference targeting PRR11 could exert anti-cancer effects, forecasting the potential application of PRR11 for targeted therapy [9,41]. In the meanwhile, circRNA_cZNF292 was involved in angiogenesis of glioma through PRR11 [77]. Notably, some biological molecules were also involved in PRR11-related targeted therapies. USP34 facilitated pancreatic cancer cell progression targeting PRR11 [55]. Surprisingly, PRR11 also acted as a significant part in a limited number of clinical cases. Chalcone, modificated by dithiocarbamate scaffolds, exert inhibitory effects via PRR11 downregulation [30]. Furthermore, PRR11 was involved in the treatment of ultrasonic irradiation for breast cancer [41]. Therefore, targeting PRR11 might provide a reliable and promising therapy for human malignancies (Figure 3).

## 6. Conclusions and Future Prospective

Concluding from the experimental and clinical data reviewed above, PRR11 is involved in cellular activities via several signaling pathways and biological molecules, manifesting its potential application in the targeted therapies. As for its contribution in tumorigenicity, PRR11 facilitates cellular proliferation in vitro in BRCA, Pancreatic cancer and Osteosarcoma; likewise, it contributes to the tumorigenicity of in vitro and in vivo models in TSCC, ESCC, NSCLC, GC, CRC, HCC, HCCA and Ovarian cancer. In considering clinical characteristics of patients with various solid tumors, PRR11 is more or less significantly correlated with greater tumor size, higher Ki-67, advanced clinical stage, more lymph node metastasis, more tumor differentiation and higher recurrence. Hence, PRR11 serves as a reliable prognostic and diagnostic indicator in TSCC, NSCLC, BRCA, GC, HCC, HCCA, Pancreatic cancer and Ovarian cancer. Moreover, traumatic autophagy, instead of protective autophagy, is notably reduced by PRR11 in cancer cells. Though PRR11 expression is downregulated in LAML patients from the GEO database, the oncogenic role of PRR11 in a variety of human cancers have already been verified. Therefore, PRR11 exerts oncogenic effects on several human cancers, which could be applied into clinical cancer treatment.

However, the existing research data of PRR11 is still not integrated. For instance, the clinical significance and cellular activities of the human cancers mentioned above are more or less in deficiency, thus clinical application would encounter great difficulties. Moreover, in gastric cancer stem cells, Hu et al. firstly applied dorsomorphin, one of the specific inhibitors of MAPK signaling, to just simply explore its involvement with PRR11 activities [6]. Conversely, our reviewed results focus on a limited number of human cancers, appealing to the reliable evaluation for PRR11 expression in other tumors. Univariate Cox regression analysis of the cancer genome atlas (TCGA) and GEO databases might be an option [78]. Additionally, the significance of clinical samples is testified [29]. Gu et al. also verified the feasibility of weighted gene co-expression network analysis (WGCNA) [49]. Hence, there is an urgent demand to refine the current data and probe the role of PRR11 in other tumors.

Of course, there are still some preliminary explorations in the correlation of PRR11 and upstream or downstream targets, further promoting its application in targeted therapy. The MirTarget program can discern the number of miRNAs binding sites with the genes [79]. The technology of the protein chip system could also be utilized for screening upstream or downstream targets of PRR11 [80]. Furthermore, functional gain or loss experiments might be reliable options [1,10].

All the above findings reveal that PRR11 serves as an oncogenic factor in cellular proliferation, migration, invasion, cell-cycle progression, apoptosis and autophagy in human cancers, including TSCC, ESCC, NSCLC, BRCA, GC, CRC, HCC, HCCA, Pancreatic cancer, Ovarian cancer and Osteosarcoma. Moreover, PRR11 positively correlates with clinical stage, tumor size, lymph node metastasis and tumor differentiation, leading to poor overall survival. Further mechanical investigation demonstrates that EMT, PI3K/AKT/mTOR, Wnt/β-catenin signalings and biological molecules, including MMP2, TIMP-2, p21, p27, CDK2, cyclin A2, CTHRC1, LXN, EGR1, UCHL1, SNAT1 and PTTG1, are involved in the functional process of PRR11, providing potential clinical applications of PRR11 for targeted therapies. While the existing data is still far from satisfactory, further investigation should be performed for the verification of the oncogenic role of PRR11.

## Figures and Tables

**Figure 1 biomolecules-12-01800-f001:**
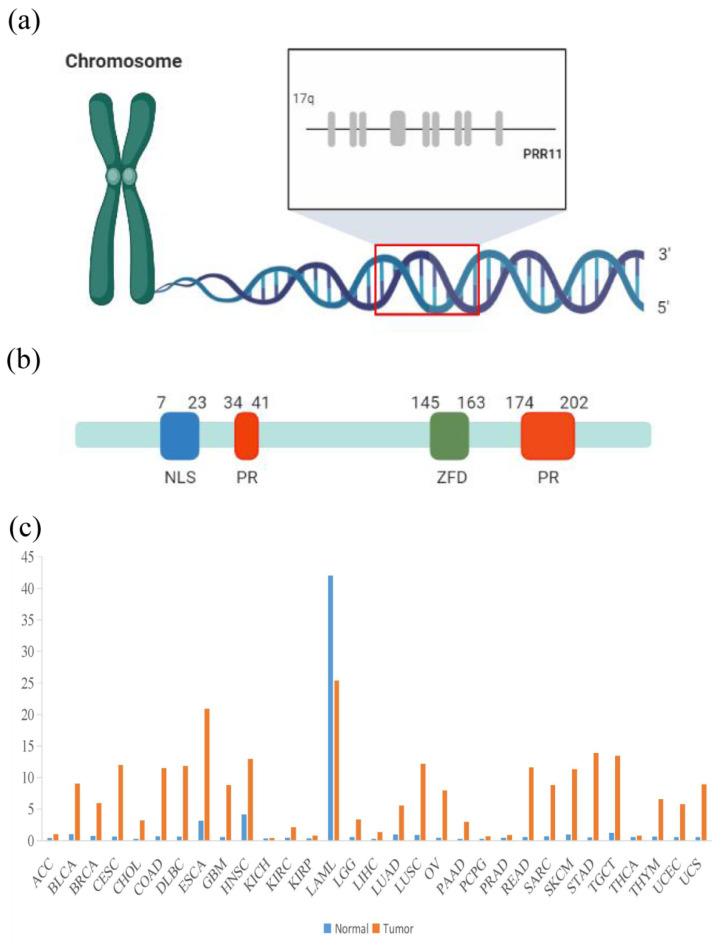
(**a**) Genetic phenotype of PRR11. PRR11 is situated in chromosome 17q22-23 with ten exons and nine introns. (**b**) Proteomics of PRR11. PRR11 consists of a nuclear localization signal (NLS), two proline rich regions (PRs) and a zinc finger domain (ZFD). (**c**) The expression level of PRR11 in multiple human cancers in the GEO database. Abbreviations: *PRR11*, proline rich 11; *NLS*, nuclear localization sequence; *PR*, proline rich region; *ZFD*, zinc finger domain; *ACC*, adrenocortical carcinoma; *BLCA*, bladder carcinoma; *BRCA*, breast carcinoma; *CESC*, cervical squamous cell carcinoma; *CHOL*, cholangiocarcinoma; *COAD*, colon adenocarcinoma; *DLBC*, diffuse large B-cell lymphoma; *ESCA*, esophageal carcinoma; *GBM*, glioblastoma; *HNSC*, head and neck squamous cell carcinoma; *KICH*, kidney chromophobe; *KIRC*, kidney renal clear cell carcinoma; *KIRP*, kidney renal papillary cell carcinoma; *LAML*, acute myeloid leukemia; *LGG*, low grade glioma; *LIHC*, liver hepatocellular carcinoma; *LUAD*, lung adenocarcinoma; *LUSC*, lung squamous cell carcinoma; *OV*, ovarian serous cystadenocarcinoma; *PAAD*, pancreatic adenocarcinoma; *PCPG*, pheochromocytoma and paraganglioma; *PRAD*, prostate adenocarcinoma; *READ*, rectum adenocarcinoma; *SARC*, sarcoma; *SKCM*, skin cutaneous melanoma; *STAD*, stomach adenocarcinoma; *TGCT*, testicular germ cell tumors; *THCA*, thyroid carcinoma; *THYM*, thymoma; *UCEC*, uterine corpus endometrial carcinoma; *UCS*, uterine carcinosarcoma.

**Figure 2 biomolecules-12-01800-f002:**
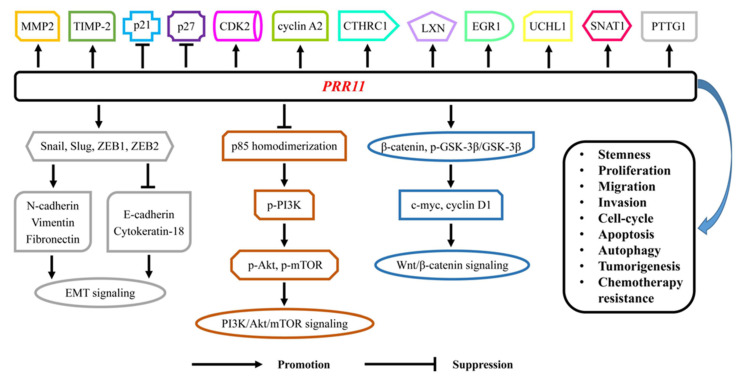
PRR11 involved oncogenic signaling pathways and biological molecules in cancers. EMT, PI3K/AKT/mTOR, Wnt/β-catenin signalings and biological molecules, including MMP2, TIMP-2, p21, p27, CDK2, cyclin A2, CTHRC1, LXN, EGR1, UCHL1, SNAT1 and PTTG1, were involved in the functional process of PRR11, including stemness, proliferation, migration, invasion, cell-cycle, apoptosis, autophagy, tumorigenesis and chemotherapy resistance. Abbreviations: *PRR11*, proline rich 11; *MMP2*, matrix metallopeptidase 2; *TIMP-2*, tissue inhibitor of metalloproteases 2; *CDK2*, cyclin-dependent kinase 2; *CTHRC1*, collagen triple helix repeat containing 1; *LXN*, latexin; *EGR1*, early growth response protein 1; *UCHL1*, ubiquitin carboxyl-terminal hydrolase 1; *SNAT1*, system a amino acid transporter 1; *ZEB1*, zinc finger E-box binding homeobox 1.

**Figure 3 biomolecules-12-01800-f003:**
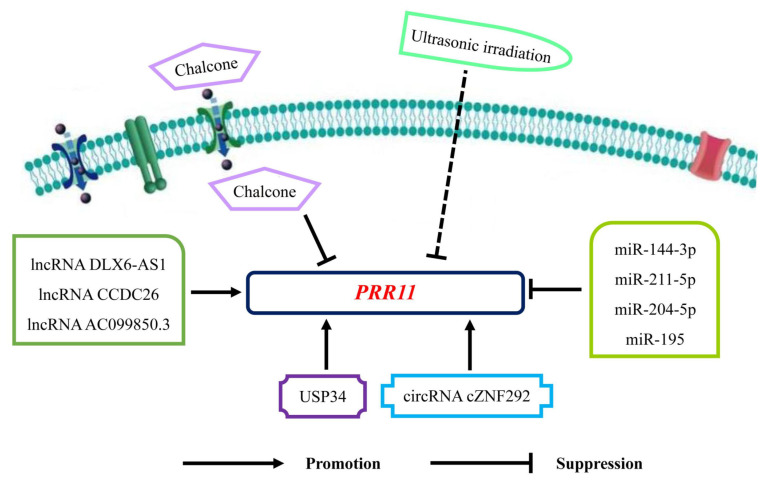
PRR11 involved targeted therapies in various malignancies. LncRNA DLX6-AS1, CCDC26 and AC099850.3 modulate PRR11 expression by functioning as competing endogenous RNAs (ceRNAs). MiRNAs, including miR-144-3p, miR-211-5p, miR-204-5p and miR-195, regulate PRR11 expression targeting PRR11 3′-UTR. Other biological molecules, such as USP34 and circRNA_cZNF292, are also involved in modulation of PRR11 expression. Moreover, Chalcone and ultrasonic irradiation could downregulate PRR11 expression to suppress cancer development. Abbreviations: *PRR11*, proline rich 11; *USP34*, ubiquitin specific protease 34.

**Table 1 biomolecules-12-01800-t001:** Biological activities of PRR11 in cancers.

Human Cancers	Assessed Cell Lines	Targets	Related Biological Activities	Ref
Tongue squamous cell carcinoma (TSCC)	SCC15, HSC3	p21, p27, CDK2, cyclin A	↑proliferation, ↑invasion, ↑cell-cycle progression, ↑tumorigenicity	[8]
Esophageal squamous cell carcinoma (ESCC)	EC9706, EC109	Akt, EMT signaling	↑proliferation, ↑migration, ↑invasion	[4]
	KYSE30, ECA109	Wnt/β-catenin signaling	↑stem cell-like traits, ↑tumorigenicity	[15]
Non-small cell lung cancer (NSCLC)	H1299	\	↑proliferation, ↑migration, ↑cell-cycle progression, ↑invasion, ↑tumorigenicity	[2]
	H1299, A549	\	↑proliferation, ↑cell-cycle progression, ↑tumorigenicity	[9]
	H1299, A549	Akt/mTOR signaling	↑proliferation, ↓apoptosis, ↓autophagy	[10]
	NCI-H460	E2F1/PTTG1	↑proliferation, ↑migration	[16]
Breast cancer (BRCA)	MCF7, HCC1428	p85, PI3K signaling	↑proliferation, ↑antiestrogen resistance	[1]
	HS578T, MDA-MB-231	EMT signaling	↑proliferation, ↑invasion	[5]
	MCF7, MDA-MB-231	\	↑proliferation, ↑migration, ↑invasion	[17]
	AU565, MDA-MB-231	\	↑proliferation, ↑migration, ↑cell-cycle progression, ↑invasion	[18]
Gastric cancer (GC)	SGC-7901, HGC-27	MAPK signaling	↑self-renewal, ↑stemness, ↑tumorigenicity	[6]
	SGC-7901	CTHRC1, LXN	↑proliferation, ↑colony formation, ↑tumorigenicity	[13]
Colorectal cancer (CRC)	SW480, HCT116	EGFR/ERK/AKT signaling CTHRC1	↑proliferation, ↑migration, ↑invasion, ↑xenograft tumor growth	[19]
Hepatocellular carcinoma (HCC)	HepG2	E2F1/PTTG1	↑proliferation, ↑migration	[16]
	Huh7	EMT, β-catenin signaling	↑proliferation, ↑migration, ↑invasion, ↑xenograft tumor growth	[20]
Hilar cholangiocarcinoma (HCCA)	QBC939	EMT signalingUCHL1, SNAT1, EGR1	↑proliferation, ↑migration, ↑cell-cycle progression, ↑xenograft tumor growth	[12]
Pancreatic cancer	Capan-1	\	↑proliferation, ↑migration	[14]
	BxPC3	E2F1/PTTG1	↑proliferation, ↑migration	[16]
Ovarian cancer	Caov3, HO-8910	PI3K/AKT/β-catenin signaling, c-myc, cyclin D1, MMP2, TIMP-2	↑proliferation, ↑migration, ↑invasion	[7]
	SKOV3, HO-8910	EGR1, N-cadherin	↑proliferation, ↑migration, ↑xenograft tumor growth	[11]
Osteosarcoma	SAOS2, MG63, U2OS	Wnt/β-catenin, EMT signaling	↑proliferation, ↑migration, ↑invasion, ↓apoptosis	[21]

Abbreviations: *CDK2*, cyclin-dependent kinase 2; *PTTG1*, pituitary tumor-transforming gene 1; *CTHRC1*, collagen triple helix repeat containing 1; *LXN*, latexin; *UCHL1*, ubiquitin carboxyl-terminal hydrolase 1; *SNAT1*, system a amino acid transporter 1; *EGR1*, early growth response protein 1; *MMP2*, matrix metallopeptidase 2; *TIMP-2*, tissue inhibitor of metalloproteases 2.

**Table 2 biomolecules-12-01800-t002:** Clinical significance of PRR11 in cancers.

Human Cancers	Clinical Cases or Tissues	Clinical Significance	Ref
Tongue squamous cell carcinoma (TSCC)	126 TSCC and 12 non-cancerous tongue tissue samples	↑clinical stage, ↑T classification, ↑N classification, ↓vital status, ↓survival time	[22]
Non-small cell lung cancer (NSCLC)	40 lung cancer tissues, 8 normal tissues and 246 lung cancer patients	↑clinical stage, ↓overall survival	[2,9]
Breast cancer (BRCA)	260 ER^+^ BRCA patients	↓recurrence free survival, ↓relapse-free survival (RFS), ↑Ki67, ↑endocrine resistance	[1]
	109 BRCA patients	↑lymph node metastasis, ↓overall survival	[5]
Gastric cancer (GC)	216 GC patients	↑T stage, ↑TNM stage, ↑tumor differentiation, ↓overall survival	[13]
Hepatocellular carcinoma (HCC)	80 pairs of HCC tissues and adjacent non-tumor liver tissues	↑tumor size, ↑TNM stage, ↓overall survival	[20]
Hilar cholangiocarcinoma (HCCA)	49 HC patients	↑invasion, ↑lymph node metastasis, ↑CA199, ↑recurrence, ↓disease-free survival	[12]
Pancreatic cancer	38 pancreatic cancer samples and 10 normal pancreatic tissues	↑invasion, ↑tumor differentiation, ↓overall survival	[14]
Ovarian cancer	51 pairs of OC tissues and normal ovarian surface epithelium tissues	↑FIGO stage, ↑tumor size, ↑lymph node metastasis	[7]
	49 primary invasive OC patients	↑FIGO stage, ↓overall survival	[11]
Osteosarcoma	62 pairs of osteosarcoma tissues and adjacent non-tumor tissues	↑tumor size, ↑Enneking stage, ↑lymph node metastasis	[21]

Abbreviations: *TNM*, tumor node metastasis; *FIGO*, federation international of gynecology and obstetrics.

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
