# Peer review of "PRR11 in Malignancies: Biological Activities and Targeted Therapies"

_biomolecules, 2022, doi:10.3390/biom12121800_

Round 1

Reviewer 1 Report

Prof Dr/Editor, thank you for inviting me to review this interesting manuscript. I have the following comments:

Line 25: Please, mention the database you used to generate the Figure 1C in the Figure title.

Line 278: Please, change the word ‘’discussion’’ to ‘’conclusion’’.

Please, add a separate title exploring the interaction of PRR11 and miRNA as stated in the following articles:

1.      MicroRNA-26b-5p suppresses the proliferation of tongue squamous cell carcinoma via targeting proline rich 11 (PRR11).

2.      miR-195 inhibits cell proliferation and angiogenesis in human prostate cancer by downregulating PRR11 expression.

3.      MicroRNA-211-5p promotes apoptosis and inhibits the migration of osteosarcoma cells by targeting proline-rich protein PRR11.

4.      miR-204-5p Hampers Breast Cancer Malignancy and Affects the Cell Cycle by Targeting PRR11.

5.      miR-144-3p Induces Cell Cycle Arrest and Apoptosis in Pancreatic Cancer Cells by Targeting Proline-Rich Protein 11 Expression via the Mitogen-Activated Protein Kinase Signaling Pathway.

Please, add a separate title exploring the interaction of PRR11 and LncRNA as stated in the following articles:

1.      LncRNA AC099850.3 promotes hepatocellular carcinoma proliferation and invasion through PRR11/PI3K/AKT axis and is associated with patients prognosis.

2.      Knockdown of lncRNA DLX6-AS1 inhibits cell proliferation, migration and invasion while promotes apoptosis by downregulating PRR11 expression and upregulating miR-144 in non-small cell lung cancer.

Author Response

1.Line 25: Please, mention the database you used to generate the Figure 1C in the Figure title.

[Response] Thank you! We have mentioned the database used to generate the Fig.1C in Line 31-32, which also has been added in the notes of Fig.1 in Line 29. Please check!

2.Line 278: Please, change the word ‘’discussion’’ to ‘’conclusion’’.

[Response] Thank you! We have changed the word accordingly.

3.Please, add a separate title exploring the interaction of PRR11 and miRNA as stated in the following articles:

1.MicroRNA-26b-5p suppresses the proliferation of tongue squamous cell carcinoma via targeting proline rich 11 (PRR11).

2.miR-195 inhibits cell proliferation and angiogenesis in human prostate cancer by downregulating PRR11 expression.

3.MicroRNA-211-5p promotes apoptosis and inhibits the migration of osteosarcoma cells by targeting proline-rich protein PRR11.

4.miR-204-5p Hampers Breast Cancer Malignancy and Affects the Cell Cycle by Targeting PRR11.

5.miR-144-3p Induces Cell Cycle Arrest and Apoptosis in Pancreatic Cancer Cells by Targeting Proline-Rich Protein 11 Expression via the Mitogen-Activated Protein Kinase Signaling Pathway.

[Response] Thanks for your constructive suggestions! We have added a separate title exploring the interaction of PRR11 and miRNA in Line 262-272. Please check!

4.Please, add a separate title exploring the interaction of PRR11 and LncRNA as stated in the following articles:

1.LncRNA AC099850.3 promotes hepatocellular carcinoma proliferation and invasion through PRR11/PI3K/AKT axis and is associated with patients prognosis.

2.Knockdown of lncRNA DLX6-AS1 inhibits cell proliferation, migration and invasion while promotes apoptosis by downregulating PRR11 expression and upregulating miR-144 in non-small cell lung cancer.

[Response] Thanks for your suggestive comments! We have added a separate title exploring the interaction of PRR11 and lncRNA in Line 273-280. Please check!

Reviewer 2 Report

The authors reviewed the role of PRR11 in malignancies, but some important aspects are missed and the authors are required to add these issues to the manuscript including:

- The contribution of PRR11 in tumorigenecity (in vitro and in vivo models)

- The importance of PRR11 in patients' survival 

- Prognostic and diagnostic value of PRR11 in different malignancies

- A schematic view of PRR11 situation in crosstalk with different cell signaling pathways and biomolecules describing PRR11 role as a major or minor component.

Author Response

- The contribution of PRR11 in tumorigenecity (in vitro and in vivo models)

[Response] Thanks for your constructive suggestions! We have added some sentences concerning the contribution of PRR11 in tumorigenecity (in vitro and in vivo models) in Line 319-327. Please check!

- The importance of PRR11 in patients' survival

[Response] Thanks for your constructive suggestions! We have added some sentences concerning the importance of PRR11 in patients' survival in Line 319-327. Please check!

- Prognostic and diagnostic value of PRR11 in different malignancies

[Response] Thanks for your constructive suggestions! We have added some sentences concerning prognostic and diagnostic value of PRR11 in different malignancies in Line 319-327. Please check!

- A schematic view of PRR11 situation in crosstalk with different cell signaling pathways and biomolecules describing PRR11 role as a major or minor component.

[Response] Thank you! We have added some sentences concerning the contribution of PRR11 in tumorigenecity (in vitro and in vivo models) in Line 230-255, 319-327. Please check!

Round 2

Reviewer 2 Report

I cannot detect the changes that authors referred in lines 319-327.

Author Response

- The contribution of PRR11 in tumorigenecity (in vitro and in vivo models)

[Response] Thanks for your constructive suggestions! We have added some sentences “As for its contribution in tumorigenecity, PRR11 facilitated cellular proliferation in vitro in BRCA, Pancreatic cancer and Osteosarcoma, and tumorigenecity in vitro and in vivo models in TSCC, ESCC, NSCLC, GC, CRC, HCC, HCCA and Ovarian cancer.”concerning the contribution of PRR11 in tumorigenecity (in vitro and in vivo models) in Section 5 Line 326-333. Please check!

- The importance of PRR11 in patients' survival

[Response] Thanks for your constructive suggestions! We have added some sentences “Considering clinical characteristics of patients with various solid tumors, PRR11, more or less, were significantly correlated with greater tumor size, higher Ki-67, advanced clinical stage, more lymph node metastasis, more tumor differentiation and higher recurrence. Hence, PRR11 served as a reliable prognostic and diagnostic indicator in TSCC, NSCLC, BRCA, GC, HCC, HCCA, Pancreatic cancer and Ovarian cancer.” concerning the importance of PRR11 in patients' survival in Line 328-333. Please check!

- Prognostic and diagnostic value of PRR11 in different malignancies

[Response] Thanks for your constructive suggestions! We have added some sentences “Considering clinical characteristics of patients with various solid tumors, PRR11, more or less, were significantly correlated with greater tumor size, higher Ki-67, advanced clinical stage, more lymph node metastasis, more tumor differentiation and higher recurrence. Hence, PRR11 served as a reliable prognostic and diagnostic indicator in TSCC, NSCLC, BRCA, GC, HCC, HCCA, Pancreatic cancer and Ovarian cancer.” concerning prognostic and diagnostic value of PRR11 in different malignancies in Line 328-333. Please check!

- A schematic view of PRR11 situation in crosstalk with different cell signaling pathways and biomolecules describing PRR11 role as a major or minor component.

[Response] Thank you! We have elaborated the crosstalk between different cell signaling pathways, biomolecules and PRR11 in Line 232-257 and added figure legends of Fig.2-3 in Line 47-49, 320-321. And we also have further discussed the interactions between PRR11 and miRNAs, lncRNAs and other molecules. The sentences “In recent years, targeted therapies gradually exhibit potential application in cancer treatment, with extraordinary clinical efficacy [66, 67]. Based on these investigational results, targeted therapies of PRR11,especially interactions between PRR11 and miRNAs, lncRNAs and other molecules, would be utilized for cancer treatment. 

4.1 Interactions between PRR11 and miRNAs

miRNAs, a group of non-coding RNAs, play vital roles in the initiation and progression of human cancers [68]. In most circumstances, miRNAs induce degradation and translational repression of target mRNAs to modulate gene expression targeting the 3' untranslated region (3' UTR) of mRNAs [69]. For instance, the luciferase activity of the PRR11 3-UTR wild-type reporter gene, not PRR11 3-UTR mutated reporter gene, was significantly decreased by miRNA-26b-5p, thus suppressing PRR11 expression [70]. Similarly in prostate cancer, osteosarcoma, pancreatic cancer and breast cancer, miR-195, miRNA-211-5p, miR-204-5p and miR-144-3p directly targeted PRR11 3'-UTR to participate in cellular activities, including proliferation, migration, cell cycle, apoptosis and angiogenesis [37, 55, 71, 72]. Therefore, exploring more miRNAs targeting PRR11 3'-UTR might be reliable approaches to regulate PRR11 expression in cancer treatment.

4.2 Interactions between PRR11 and lncRNAs

LncRNAs usually modulate gene expression by functioning as competing endogenous RNAs (ceRNAs), which sequester miRNAs and therefore protect their target mRNAs from repression [73]. As mentioned above, lncRNA DLX6-AS1 serve as a ceRNA of miR-144 in NSCLC cells to upregulate PRR11 expression [32]. In particular, lncRNA CCDC26 elevated circRNA_ANKIB1 expression, acting as a ceRNA of miR-195-5p, to upregulate PRR11 expression in myeloid leukemia cells [74]. However, the biological mechanism of lncRNA AC099850.3 as an oncogene in hepatocellular carcinoma via PRR11/PI3K/AKT axis is still uncertain [75].

4.3 Interactions between PRR11 and other molecules

Preliminary experiments manifested that SonoVue microbubbles-mediated RNA interference targeting PRR11 could exert anti-cancer effects, forecasting the potential application of PRR11 for targeted therapy [9, 38]. In the meanwhile, circRNA_cZNF292 was involved in angiogenesis of glioma through PRR11 [76]. ”have been added in section 4, Line 260-286. Please check!

Round 3

Reviewer 2 Report

While the authors discussed different aspects of PRR11 in the manuscript, the suggested topics including involvement of PRR11 in tumorigenesis, cancer prognosis and diagnosis, as well as patients survival (which are most important sub-titles of the manuscript title) are too shortly discussed in current form and should be more extended.

Author Response

While the authors discussed different aspects of PRR11 in the manuscript, the suggested topics including involvement of PRR11 in tumorigenesis, cancer prognosis and diagnosis, as well as patients survival (which are most important sub-titles of the manuscript title) are too shortly discussed in current form and should be more extended.

[Response] Thanks for your constructive suggestions! We have added a separate topic named 4. Biological and clinical significance of PRR11 in human cancers to elucidate the involvement of PRR11 in tumorigenesis, cancer prognosis and diagnosis, as well as patients survival. The sentences 4.1 PRR11 involved in tumorigenesis of human cancers-Cancer stem cells, considered as a vital cell cluster in cellular proliferation, drug resistance and tumor recurrence, are equipped with self-renewal and multi-directional differentiation [66]. In ESCC, BRCA and GC, PRR11 facilitated self-renewal and stemness of stem cells, contributing drug resistance [1, 6, 24]. Subsequently, in vitro and in vivo models were utilized for evaluation of PRR11 in tumorigenesis. In vitro, cellular proliferation and cell-cycle progression were evidently boosted by PRR11 in TSCC, ESCC, NSCLC, BRCA, GC, CRC, HCC, HCCA, Pancreatic cancer, Ovarian cancer and Osteosarcoma [4, 5, 7, 8, 10, 12-14, 31, 43, 61]. Similarly in vivo models, PRR11 promoted xenograft tumor growth in tumor size, weight and Ki-67 expression in TSCC, NSCLC, GC, CRC, HCC and Ovarian cancer [2, 6, 8, 9, 11, 13, 43, 48]. Hence, PRR11 might be great of biological significance in tumorigenesis of human cancers.

4.2 PRR11 involved in diagnosis and prognosis of malignancies-Compared with non-tumorous ones, PRR11 was evidently upregulated in tumor tissues, including ESCC, NSCLC, CRC, HCC, Pancreatic cancer, Ovarian cancer and Osteosarcoma, which would contribute to improvement of clinical diagnosis in these cancers [2, 4, 7, 9, 14, 23, 28, 43, 48, 61]. More importantly, PRR11 also served as a prognostic biomarker for several human cancers. Considering clinical features and prognosis of these patients, PRR11 were positively correlated with earlier tumor invasion, advanced clinical stage, TNM classification, more lymph node metastasis and higher Ki-67 ratio, which usually were accompanied by shorter overall survival in TSCC, NSCLC, BRCA, GC, HCC, Pancreatic cancer, Ovarian cancer and Osteosarcoma [1, 5, 7, 11-14, 18, 35, 48, 53, 61]. Therefore, PRR11 serves as a promising prognostic indicator in a limited number of human cancers, gradually manifesting its potential application for targeted therapies (Tab.1-2). were added in Line 259-280. Please check!